# A Mixture Autoregressive Model Based on an Asymmetric Exponential Power Distribution

**Yunlu Jiang * and Zehong Zhuang**

Department of Statistics, College of Economics, Jinan University, Guangzhou 510632, China
* Correspondence: tjiangyl@jnu.edu.cn

**Abstract:** In nonlinear time series analysis, the mixture autoregressive model (MAR) is an effective statistical tool to capture the multimodality of data. However, the traditional methods usually need to assume that the error follows a specific distribution that is not adaptive to the dataset. This paper proposes a mixture autoregressive model via an asymmetric exponential power distribution, which includes normal distribution, skew-normal distribution, generalized error distribution, Laplace distribution, asymmetric Laplace distribution, and uniform distribution as special cases. Therefore, the proposed method can be seen as a generalization of some existing model, which can adapt to unknown error structures to improve prediction accuracy, even in the case of fat tail and asymmetry. In addition, an expectation-maximization algorithm is applied to implement the proposed optimization problem. The finite sample performance of the proposed approach is illustrated via some numerical simulations. Finally, we apply the proposed methodology to analyze the daily return series of the Hong Kong Hang Seng Index. The results indicate that the proposed method is more robust and adaptive to the error distributions than other existing methods.

**Keywords:** mixture autoregressive (AR) models; AEP density; EM algorithm

**MSC:** 62G35; 62H30

## 1. Introduction

In the study of time series, autoregressive (AR) models are a fundamental and important statistical tool. The classical AR models only allow for unimodal marginal and conditional densities, which cannot capture conditional heteroscedasticity. To solve this problem, Wong & Li (2000) introduced the *k*-component Gaussian mixture AR (GMAR) model that is presented as follows [1].

Let $X_t$ be a random variable observed at time $t$, and let $\mathcal{F}_t$ be the information set up to time $t$. $X_t$ arises from a *k*-component GMAR model of order *p* if $X_t|\mathcal{F}_{t-1}$ has a density of the form

$$g(x_t|\mathcal{F}_{t-1};\gamma) = \sum_{i=1}^{k} \pi_i \phi\left(x_t; \beta_{i0} + \sum_{j=1}^{p} \beta_{ij} x_{t-j}, \sigma_i^2\right), \tag{1}$$

where $\pi_i > 0$, $\sum_{i=1}^{k} \pi_i = 1$, $\boldsymbol{\beta}_i = (\beta_{i0}, \cdots, \beta_{ip})^\top$, $\sigma_i^2 > 0$ for all $i = 1, \cdots, k$, $\gamma = (\pi_1, \boldsymbol{\beta}_1^\top, \sigma_1^2, \cdots, \pi_k, \boldsymbol{\beta}_k^\top, \sigma_k^2)^\top$ is an unknown parameter vector, and $\phi(x; \mu, \sigma^2)$ is a normal density function with mean $\mu$ and variance $\sigma^2$.

The GMAR model (1) is very useful for modeling nonlinear time series, and it can capture serial correlations, time-varying means, and volatilities [2]. Furthermore, Wong & Li (2001) and Fong et al. (2007) extended the GMAR model to the AR conditional heteroscedastic (ARCH) model and the vector AR model, respectively [3,4]. Since the GMAR model needs the Gaussian assumption, its estimator is not robust to heavy-tailed data or outliers. In order to estimate the occurrence of extreme financial events accurately, Wong et al. (2009)

proposed a Student *t*-mixture AR model [2]. Nguyen et al. (2016) introduced the Laplace mixture AR model [5]. Meitz et al. (2021) considered a mixture of autoregressive models based on the scale mixture of skew-normal distributions [6]. Meitz et al. (2021) proposed a new mixture autoregressive model based on Student's t-distribution [7]. Virolainen (2021) introduced a new mixture autoregressive model that combines Gaussian and Student's *t* mixture components [8]. Solikhah et al. (2021) studied Fisher's z distribution-based mixture autoregressive model [9]. Since the above proposed methods need to assume a specific error distribution, they are not adaptive to the error distributions. A wrong distributional assumption may lead to a decrease in the precision of the model estimate.

In order to develop a robust mixture autoregressive, in this paper, we proposed a robust estimation procedure for mixture AR models by replacing a normal density function in (1) with an asymmetric exponential power (AEP) density function [10]. The AEP distribution includes many important statistical distributions as special cases, e.g., normal distribution, skew-normal distribution, generalized error distribution, Laplace distribution, asymmetric Laplace distribution, and uniform distribution. This indicates that the proposed method provides a more general approach, which can adapt to much more different error structures and automatically chooses the parameters to achieve both efficiency and robustness of estimators. Meanwhile, we apply an expectation-maximization (EM) algorithm [11] to implement the proposed optimization problem. In addition, the finite-sample performance of the proposed method is evaluated via some numerical studies and a real-data analysis.

The remainder of the paper is organized as follows. In Section 2, we introduce an estimation procedure for mixture AR models via an AEP density function and introduce an EM algorithm to solve the proposed methodology. In Section 3, simulation studies are conducted to evaluate the finite sample performance of the proposed method. In Section 4, a real data set is analyzed to compare the proposed method with some existing methods. We conclude with some remarks in Section 5.

## 2. Methodology

According to Fernandez et al. (1995), an AEP density $f(x; \mu, \tau, \alpha, \sigma)$ is defined as follows [10].

$$f(x; \mu, \sigma, \alpha, \tau) = \frac{\alpha \tau (1-\tau)}{\Gamma(1/\alpha)\sigma} \exp\left\{-\frac{|x-\mu|^\alpha}{\sigma^\alpha}\right\} [I(x \geq \mu)\tau^\alpha + I(x < \mu)(1-\tau)^\alpha], \quad (2)$$

where $\mu \in \mathbb{R}$ is the location parameter, $\sigma > 0$ is the scale parameter, $0 < \tau < 1$ controls the skewness, $\alpha > 0$ is the shape parameter, and $I(\cdot)$ is an indicator function. The AEP density function is a flexible and general density function class that can even capture the fat tail and asymmetry of the error term. It also includes some important statistical density functions as its special cases, e.g.,

1. Normal density function: $f(x; \mu, \sigma, \alpha = 2, \tau = 0.5)$.
2. Skew-normal density function: $f(x; \mu, \sigma, \alpha = 2, \tau)$.
3. Generalized error density function: $f(x; \mu, \sigma, \alpha, \tau = 0.5)$.
4. Laplace density function: $f(x; \mu, \sigma, \alpha = 1, \tau = 0.5)$.
5. Asymmetric Laplace density function: $f(x; \mu, \sigma, \alpha = 1, \tau)$.
6. Uniform density function: $f(x; \mu, \sigma, \alpha \to \infty, \tau)$.

Based on the AEP density function, we propose the *k*-component AEP-MAR model, which is defined as follows:

$$h(x_t | \mathcal{F}_{t-1}; \boldsymbol{\theta}) = \sum_{i=1}^{k} \pi_i f_i\left(x_t; \beta_{i0} + \sum_{j=1}^{p} \beta_{ij} x_{t-j}, \sigma_i, \alpha_i, \tau_i\right),$$

where $\boldsymbol{\theta} = (\pi_1, \boldsymbol{\beta}_1^\top, \sigma_1, \alpha_1, \tau_1, \cdots, \pi_k, \boldsymbol{\beta}_k^\top, \sigma_k, \alpha_k, \tau_k)^\top$ is an unknown parameter vector, and $f_i$ is an AEP density function given in (2). For the *k*-component AEP-MAR model, we obtain the conditional expectation and conditional variance as follows:

$$E(x_t|\mathcal{F}_{t-1};\boldsymbol{\theta}) = \sum_{i=1}^{k} \pi_i \left( \beta_{i0} + \sum_{j=1}^{p} \beta_{ij} x_{t-j} \right) = \sum_{i=1}^{k} \pi_i \mu_i,$$

$$Var(x_t|\mathcal{F}_{t-1};\boldsymbol{\theta}) = \sum_{i=1}^{k} \pi_i \sigma_i^2 + \sum_{i=1}^{k} \pi_i \mu_i^2 - \left( \sum_{i=1}^{k} \pi_i \mu_i \right)^2.$$

where $\mu_i = \beta_{i0} + \sum_{j=1}^{p} \beta_{ij} x_{t-j}$.

Let $x_1, x_2, \cdots, x_n$ be a random sample from the $k$-component AEP-MAR model. Then, the sample conditional log-likelihood function can be written as

$$P_n(\boldsymbol{\theta}) = \sum_{t=p+1}^{n} \log \left\{ \sum_{i=1}^{k} \pi_i \frac{\alpha_i \tau_i (1-\tau_i)}{\Gamma(1/\alpha_i)\sigma_i} \exp\left( -\frac{|x_t - \beta_{i0} - \sum_{j=1}^{p} \beta_{ij} x_{t-j}|^{\alpha_i}}{\sigma_i^{\alpha_i}} \right) \right.$$
$$\left. \left[ I(x_t \geq \beta_{i0} + \sum_{j=1}^{p} \beta_{ij} x_{t-j}) \tau_i^{\alpha_i} + I(x_t < \beta_{i0} + \sum_{j=1}^{p} \beta_{ij} x_{t-j})(1-\tau_i)^{\alpha_i} \right] \right\}$$

Therefore, an estimator $\hat{\boldsymbol{\theta}}_n$ for $\boldsymbol{\theta}$ is defined as

$$\hat{\boldsymbol{\theta}}_n = \arg\max_{\boldsymbol{\theta}} P_n(\boldsymbol{\theta}). \tag{3}$$

Theoretically, by selecting the proper parameters of location, skewness, shape, and scale, the AEP-MAR model can select the best likelihood function via the data-driven technique. Under some special conditions, the likelihood function of the AEP-MAR model can also be equivalent to the existing statistical methods, e.g., GMAR [1] and LMAR [5]. This implies that the AEP-MAR model can provide a more general approach that does not need to assume the error distribution in advance. In addition, the proposed method can adapt to the unknown error structures to improve prediction accuracy.

*Algorithm*

The EM algorithm is a commonly used algorithm for maximum likelihood estimation in incomplete data proposed by Dempster et al. (1977). Under proper regularity conditions, the EM algorithm has ascent property and global convergence [11]. In this subsection, we will apply the EM algorithm to solve (3).

Firstly, we define the unobserved random variables

$$z_{tj} = \begin{cases} 1, \text{if sample } x_t \text{ is in the } j\text{-th component,} \\ 0, \text{otherwise.} \end{cases}$$

where $t = 1, \cdots, n$ and $j = 1, \cdots, k$. Let $\mathbf{z}_i = (z_{i1}, \cdots, z_{ik})^\top$. Then, the complete data are $\{(x_t, \mathbf{z}_t), t = 1, \cdots, n\}$. Thus, the log-likelihood function of the complete data can be obtained as follows:

$$R_n(\boldsymbol{\theta}) = \sum_{t=p+1}^{n} \sum_{i=1}^{k} z_{ti} \log \left\{ \pi_i \frac{\alpha_i \tau_i (1-\tau_i)}{\Gamma(1/\alpha_i)\sigma_i} \exp\left( -\frac{|x_t - \beta_{i0} - \sum_{j=1}^{p} \beta_{ij} x_{t-j}|^{\alpha_i}}{\sigma_i^{\alpha_i}} \right) \right.$$
$$\left. \left[ I(x_t \geq \beta_{i0} + \sum_{j=1}^{p} \beta_{ij} x_{t-j}) \tau_i^{\alpha_i} + I(x_t < \beta_{i0} + \sum_{j=1}^{p} \beta_{ij} x_{t-j})(1-\tau_i)^{\alpha_i} \right] \right\}. \tag{4}$$

In the following, we apply an EM algorithm to implement (4).

**E-step:** Given the $m$-th approximation $\hat{\boldsymbol{\theta}}_n^{(m)}$ of $\boldsymbol{\theta}$, the expectation of the latent variable $z_{ti}$ is given by

$$p_{ti}^{(m)} = E(z_{ti}|\mathcal{F}_{t-1};\hat{\boldsymbol{\theta}}_n^{(m)}) \quad = \frac{\hat{\pi}_{ni}^{(m)} f_i\left(x_t; \hat{\beta}_{ni0}^{(m)} + \sum_{j=1}^p \hat{\beta}_{nij}^{(m)} x_{t-j}, \hat{\sigma}_{ni}^{(m)}, \hat{\alpha}_{ni}^{(m)}, \hat{\tau}_{ni}^{(m)}\right)}{\sum_{i=1}^k \hat{\pi}_{ni}^{(m)} f_i\left(x_t; \hat{\beta}_{ni0}^{(m)} + \sum_{j=1}^p \hat{\beta}_{nij}^{(m)} x_{t-j}, \hat{\sigma}_{ni}^{(m)}, \hat{\alpha}_{ni}^{(m)}, \hat{\tau}_{ni}^{(m)}\right)}.$$

**M-step:** By replacing $z_{ti}$ with $p_{ti}^{(m)}$ in (4), we obtain the following objective function:

$$R_n^1(\boldsymbol{\theta}) = \sum_{t=p+1}^n \sum_{i=1}^k p_{ti}^{(m)} \log \left\{ \pi_i \frac{\alpha_i \tau_i (1-\tau_i)}{\Gamma(1/\alpha_i)\sigma_i} \exp\left( -\frac{|x_t - \beta_{i0} - \sum_{j=1}^p \beta_{ij} x_{t-j}|^{\alpha_i}}{\sigma_i^{\alpha_i}} \right) \right.$$
$$\left. \left[ I(x_t \geq \beta_{i0} + \sum_{j=1}^p \beta_{ij} x_{t-j})\tau_i^{\alpha_i} + I(x_t < \beta_{i0} + \sum_{j=1}^p \beta_{ij} x_{t-j})(1-\tau_i)^{\alpha_i} \right] \right\}. \tag{5}$$

By maximizing $R_n^1(\boldsymbol{\theta})$ about $\pi_i$, we can yield

$$\hat{\pi}_{ni}^{(m+1)} = \frac{1}{n-p} \sum_{t=p+1}^n p_{ti}^{(k)}.$$

For fixed values of $\boldsymbol{\beta}_i$ and $\alpha_i$, the values of $\sigma_i$, $\tau_i$ can be expressed by maximizing $R_n^1(\boldsymbol{\theta})$:

$$\sigma_i^{(m+1)} \quad = \quad \sigma_i(\boldsymbol{\beta}_i, \alpha_i) = \left\{ \frac{\alpha_i [e^+(\boldsymbol{\beta}_i, \alpha_i)(\tau_i)^{\alpha_i} + e^-(\boldsymbol{\beta}_i, \alpha_i)(1-\tau_i)^{\alpha_i}]}{\sum_{t=p+1}^n p_{ti}^{(k)}} \right\}^{1/\alpha_i},$$

$$\tau_i^{(m+1)} \quad = \quad \tau_i(\boldsymbol{\beta}_i, \alpha_i) = \left\{ 1 + \left[ \frac{e^+(\boldsymbol{\beta}_i, \alpha_i)}{e^-(\boldsymbol{\beta}_i, \alpha_i)} \right]^{1/(\alpha_i+1)} \right\}^{-1}.$$

where

$$e^+(\boldsymbol{\beta}_i, \alpha_i) \quad = \quad \sum_{t=p+1}^n p_{ti}|x_t - \beta_{i0} - \sum_{j=1}^p \beta_{ij} x_{t-j}|\tau_i^{\alpha_i} I(x_t \geq \beta_{i0} + \sum_{j=1}^p \beta_{ij} x_{t-j}),$$

$$e^-(\boldsymbol{\beta}_i, \alpha_i) \quad = \quad \sum_{t=p+1}^n p_{ti}|x_t - \beta_{i0} - \sum_{j=1}^p \beta_{ij} x_{t-j}|\tau_i^{\alpha_i} I(x_t < \beta_{i0} + \sum_{j=1}^p \beta_{ij} x_{t-j}).$$

By replacing $\sigma_i$ and $\tau_i$ with $\sigma_i^{(m+1)}$ and $\tau_i^{(m+1)}$ in (5), the objective function about $\{\boldsymbol{\beta}_i, \alpha_i\}$ can be written as

$$R_n^2(\boldsymbol{\beta}_i, \alpha_i) = \sum_{t=p+1}^n p_{ti}^{(k)} \left\{ \log\left( \frac{\alpha_i}{\Gamma(1/\alpha_i)} \right) - \frac{1}{\alpha_i} \log\left( \frac{\alpha_i}{\sum_{t=p+1}^n p_{ti}^{(k)}} \right) - \frac{1}{\alpha_i} \right.$$

$$\left. - \frac{1+\alpha_i}{\alpha_i} \log\left[ e^+(\boldsymbol{\beta}_i, \alpha_i)^{1/(1+\alpha_i)} + e^-(\boldsymbol{\beta}_i, \alpha_i)^{1/(1+\alpha_i)} \right] \right\}. \tag{6}$$

Therefore, the $(m+1)$-th approximation $\hat{\boldsymbol{\theta}}_n^{(m+1)}$ of $\boldsymbol{\theta}$ can be obtained by:

$$\hat{\beta}_{ni}^{(m+1)} \quad = \quad \arg\max_{\boldsymbol{\beta}_i} R_n^2(\boldsymbol{\beta}_i, \hat{\alpha}_{ni}^{(m)}), i = 1, \cdots, k,$$

$$\hat{\alpha}_{ni}^{(m+1)} \quad = \quad \arg\max_{\alpha_i} R_n^2(\hat{\boldsymbol{\beta}}_{ni}^{(m)}, \alpha_i), i = 1, \cdots, k,$$

$$\hat{\sigma}_{ni}^{(m+1)} \quad = \quad \sigma_i(\hat{\boldsymbol{\beta}}_{ni}^{(m+1)}, \hat{\alpha}_{ni}^{(m+1)}), i = 1, \cdots, k,$$

$$\hat{\tau}_{ni}^{(m+1)} \quad = \quad \tau_i(\hat{\boldsymbol{\beta}}_{ni}^{(m+1)}, \hat{\alpha}_{ni}^{(m+1)}), i = 1, \cdots, k.$$

**Remark 1.** *In order to implement the above EM algorithm, we need an initial value* $\hat{\boldsymbol{\theta}}_{ni}^{(0)} = \{\hat{\boldsymbol{\beta}}_{ni}^{(0)}, \hat{\alpha}_{ni}^{(0)}, \hat{\sigma}_{ni}^{(0)}, \hat{\tau}_{ni}^{(0)}, \hat{\pi}_{ni}^{(0)}\}, i = 1, \cdots, k$. *First, we apply the k-means clustering method to the dataset. According to [12], we obtain* $\hat{\boldsymbol{\theta}}_{ni}^{(0)}$ *as follows:*

$$
\begin{aligned}
\hat{\alpha}_{ni}^{(0)} &= 1, \\
\hat{\boldsymbol{\beta}}_{ni}^{(0)} &= \boldsymbol{\beta}_i(\hat{\tau}_{ni}^{(0)}), \\
\hat{\tau}_{ni}^{(0)} &= \arg\min_{\tau_i} \sum_{t \in O_i} \rho_{\tau_i}\left( x_t - \beta_{i0}(\tau_i) - \sum_{j=1}^{p} \beta_{ij}(\tau_i)x_{t-j} \right) \Big/ [\tau_i(1-\tau_i)], \\
\hat{\sigma}_{ni}^{(0)} &= \frac{1}{|O_i|} \sum_{t \in O_i} \rho_{\hat{\tau}_{ni}^{(0)}}\left( x_t - \hat{\beta}_{ni0}^{(0)} - \sum_{j=1}^{p} \hat{\beta}_{nij}^{(0)} x_{t-j} \right),
\end{aligned}
$$

*where $O_i$ is the random sample in the i-th category, $\rho_\tau(\cdot)$ is the quantile check loss function, and $\boldsymbol{\beta}_i(\tau) = \arg\min_{\boldsymbol{\beta}_i} \sum_{t \in O_i} \rho_\tau(x_t - \beta_{i0} - \sum_{j=1}^{p} \beta_{ij}x_{t-j})$. We can use some standard numerical software to obtain $\hat{\boldsymbol{\beta}}_{ni}^{(m+1)}$ and $\hat{\alpha}_{ni}^{(m+1)}$, e.g., quantreg, optim, and optimize in R software.*

## 3. Simulation Studies

**Example 1.** *In this example, some numerical simulations are carried out to illustrate the finite-sample performance of the proposed method. We compare the proposed method (AEP-MAR) with the following three methods: the method based on the Gaussian mixture autoregressive model (GMAR) [1], the method based on the Student t-mixture autoregressive models (TMAR) [2], and the method based on the Laplace mixture autoregressive model (LMAR) [5]. In this simulation, we consider a following two-component time series model (7).*

$$
\begin{cases}
y_t = 0.6y_{t-1} - 0.9y_{t-2} + \epsilon_1, & \text{with } \pi_1 = 0.5, \\
y_t = 0.1y_{t-1} + 0.7y_{t-2} + \epsilon_2, & \text{with } \pi_2 = 0.5.
\end{cases} \tag{7}
$$

*We generate 200 random samples from model (7) with sample sizes of $n = 250, 500$. For the error terms $\epsilon_1$ and $\epsilon_2$, in order to demonstrate that the proposed method is robust to unknown error distributions, we consider the following five scenarios:*

**Scenario 1:** *The standard normal distribution ($N(0,1)$).*
**Scenario 2:** *The standard Laplace distribution ($La(0,1)$).*
**Scenario 3:** *The t-distribution with degrees of freedom 3 ($t(3)$).*
**Scenario 4:** *A mixture of standard normal distribution $N(0,1)$ and standard Laplace distribution $La(0,1)$ ($0.5N(0,1) + 0.5La(0,1)$).*
**Scenario 5:** *The chi-square distribution ($\chi^2(3)$). When the error assumption is correct, the corresponding mixture AR model should have the best performance. Meanwhile, when the error assumption is wrong, the estimation accuracy of this method will also decrease. Therefore, the Scenarios 1–3 are used to compare the performance with the existing methods under correct error assumption, and the performance of the proposed method should be similar to the correct model. The Scenarios 4 and 5 are used to demonstrate that our method is robust to unknown err structures, and the performance of the proposed method should rank first among the four methods.*

To assess the finite-sample performance, we calculate the bias and the mean squared error (MSE) of estimators based on 200 simulations. The simulations results are reported in Tables 1–7, respectively. In Tables 2 and 4, we also report the estimators of other parameters for the two-component AEP-MAR model. From Table 1, we find that the GMAR has smaller bias and MSE than other three methods in the case of normal distribution, while the finite-sample performance of the AEP-MAR is better than the other two methods. In Table 3, the LMAR has the best performance in the case of laplace distribution. Meanwhile, the performance of the AEP-MAR is also similar to that of the LMAR. We can observe from Table 5 that the TMAR has the smallest bias and MSE in all four methods, while the AEP-

MAR and the TMAR have similar performance. In Tables 6 and 7, the AEP-MAR has the smallest bias and MSE in all four methods, and the effectiveness of the other three methods decreases significantly. The estimators of AEP-MAR are also precise as the sample size increases. This illustrates that AEP-MAR is robust and effective to an unknown error structure. In conclusion, the proposed method is more adaptive to the error distribution than the other three methods. If the error structure is unknown, the proposed method should be considered first.

**Table 1.** Simulation results for **Scenario 1**.

|  |  | GMAR | TMAR | LMAR | AEP-MAR |
|---|---|---|---|---|---|
| | $\beta_{11}$ | 0.0036 (0.0026) | 0.0064 (0.0031) | −0.0054 (0.0037) | **0.0034 (0.0026)** |
| | $\beta_{12}$ | **0.0023 (0.0023)** | 0.0054 (0.0030) | 0.0151 (0.0037) | 0.0030 (0.0023) |
| | $\beta_{21}$ | 0.0010 (0.0022) | 0.0008 (0.0027) | 0.0061 (0.0032) | 0.0008 (0.0024) |
| $n = 250$ | $\beta_{22}$ | **0.0004 (0.0016)** | 0.0019 (0.0018) | 0.0075 (0.0024) | 0.0015 (0.0017) |
| | $\pi_1$ | **0.0022 (0.0014)** | 0.0025 (0.0016) | 0.0024 (0.0023) | **0.0024 (0.0014)** |
| | $\pi_2$ | **−0.0022 (0.0014)** | −0.0025 (0.0016) | −0.0024 (0.0023) | **−0.0024 (0.0014)** |
| | $\beta_{11}$ | **0.0007 (0.0009)** | 0.0018 (0.0012) | −0.0056 (0.0017) | 0.0008 (0.0009) |
| | $\beta_{12}$ | **0.0031 (0.0011)** | 0.0058 (0.0013) | 0.0078 (0.0017) | 0.0031 (0.0012) |
| $n = 500$ | $\beta_{21}$ | **0.0029 (0.0013)** | 0.0040 (0.0015) | 0.0099 (0.0022) | 0.0035 (0.0014) |
| | $\beta_{22}$ | **0.0031 (0.0012)** | 0.0104 (0.0014) | 0.0129 (0.0022) | 0.0039 (0.0013) |
| | $\pi_1$ | **0.0022 (0.0013)** | 0.0029 (0.0016) | 0.0038 (0.0015) | 0.0023 (0.0014) |
| | $\pi_2$ | **−0.0022 (0.0013)** | −0.0029 (0.0016) | −0.0038 (0.0015) | −0.0023 (0.0014) |

**Table 2.** Simulation results of AEP-MAR for **Scenario 1**.

| AEP-MAR | $\sigma_1$ | $\sigma_2$ | $\tau_1$ | $\tau_2$ | $\alpha_1$ | $\alpha_2$ |
|---|---|---|---|---|---|---|
| $n = 250$ | 0.6833 (0.0108) | 0.6886 (0.0092) | 0.5031 (0.0012) | 0.4984 (0.0010) | 2.1746 (0.3438) | 2.1036 (0.3445) |
| $n = 500$ | 0.7038 (0.0056) | 0.6990 (0.0057) | 0.5003 (0.0007) | 0.5016 (0.0006) | 2.0986 (0.1625) | 2.1009 (0.1638) |

**Table 3.** Simulation results for **Scenario 2**.

|  |  | GMAR | TMAR | LMAR | AEP-MAR |
|---|---|---|---|---|---|
| | $\beta_{11}$ | 0.0061 (0.0018) | 0.0060 (0.0012) | **0.0040 (0.0011)** | 0.0046 (0.0011) |
| | $\beta_{12}$ | 0.0013 (0.0019) | −0.0012 (0.0012) | **0.0004 (0.0010)** | 0.0012 (0.0010) |
| $n = 250$ | $\beta_{21}$ | 0.0022 (0.0019) | −0.0029 (0.0016) | 0.0009 (0.0014) | −0.0006 (0.0015) |
| | $\beta_{22}$ | 0.0088 (0.0019) | 0.0039 (0.0015) | **0.0025 (0.0013)** | 0.0026 (0.0013) |
| | $\pi_1$ | 0.0014 (0.0023) | 0.0019 (0.0022) | 0.0014 (0.0021) | 0.0013 (0.0022) |
| | $\pi_2$ | −0.0014 (0.0023) | −0.0019 (0.0022) | −0.0014 (0.0021) | −0.0013 (0.0022) |
| | $\beta_{11}$ | −0.0008 (0.0008) | 0.0011 (0.0006) | **0.0006 (0.0005)** | **0.0006 (0.0005)** |
| | $\beta_{12}$ | 0.0012 (0.0006) | 0.0010 (0.0005) | **0.0007 (0.0004)** | 0.0010 (0.0004) |
| $n = 500$ | $\beta_{21}$ | 0.0054 (0.0009) | 0.0022 (0.0005) | **0.0012 (0.0005)** | 0.0020 (0.0005) |
| | $\beta_{22}$ | 0.0048 (0.0010) | 0.0023 (0.0005) | **0.0016 (0.0005)** | 0.0019 (0.0005) |
| | $\pi_1$ | 0.0023 (0.0012) | 0.0014 (0.0009) | **0.0010 (0.0009)** | 0.0014 (0.0009) |
| | $\pi_2$ | −0.0023 (0.0012) | −0.0014 (0.0009) | **−0.0010 (0.0009)** | −0.0014 (0.0009) |

**Table 4.** Simulation results of AEP-MAR for **Scenario 2**.

| AEP-MAR | $\sigma_1$ | $\sigma_2$ | $\tau_1$ | $\tau_2$ | $\alpha_1$ | $\alpha_2$ |
|---|---|---|---|---|---|---|
| $n = 250$ | 0.4823 (0.0344) | 0.4857 (0.0420) | 0.5048 (0.0015) | 0.4957 (0.0014) | 1.0618 (0.4893) | 1.0240b (0.4989) |
| $n = 500$ | 0.4985 (0.0115) | 0.5031 (0.0137) | 0.5028 (0.0009) | 0.4990 (0.0006) | 1.0173 (0.4212) | 0.9883 (0.4293) |

**Table 5.** Simulation results for **Scenario 3**.

|  |  | GMAR | TMAR | LMAR | AEP-MAR |
|---|---|---|---|---|---|
| | $\beta_{11}$ | 0.0046 (0.0017) | **0.0030 (0.0012)** | 0.0035 (0.0015) | 0.0035 (0.0014) |
| | $\beta_{12}$ | 0.0053 (0.0020) | **−0.0008 (0.0009)** | 0.0019 (0.0013) | 0.0008 (0.0012) |
| $n = 250$ | $\beta_{21}$ | 0.0078 (0.0015) | **0.0023 (0.0008)** | 0.0044 (0.0017) | 0.0045 (0.0010) |
| | $\beta_{22}$ | 0.0102 (0.0017) | **0.0025 (0.0009)** | 0.0051 (0.0013) | 0.0028 (0.0012) |
| | $\pi_1$ | 0.0035 (0.0028) | **0.0022 (0.0021)** | 0.0030 (0.0026) | 0.0026 (0.0022) |
| | $\pi_2$ | −0.0035 (0.0028) | **−0.0022 (0.0021)** | −0.0030 (0.0026) | −0.0026 (0.0022) |
| | $\beta_{11}$ | −0.0008 (0.0008) | **0.0003 (0.0005)** | −0.0016 (0.0008) | 0.0006 (0.0007) |
| | $\beta_{12}$ | 0.0129 (0.0012) | **0.0030 (0.0004)** | 0.0037 (0.0006) | 0.0033 (0.0006) |
| $n = 500$ | $\beta_{21}$ | 0.0076 (0.0010) | **0.0015 (0.0005)** | 0.0040 (0.0007) | 0.0018 (0.0006) |
| | $\beta_{22}$ | 0.0106 (0.0015) | **0.0012 (0.0004)** | 0.0040 (0.0005) | 0.0022 (0.0005) |
| | $\pi_1$ | 0.0018 (0.0024) | **0.0001 (0.0010)** | 0.0014 (0.0012) | 0.0008 (0.0011) |
| | $\pi_2$ | −0.0018 (0.0024) | **−0.0001 (0.0010)** | −0.0014 (0.0012) | −0.0008 (0.0011) |

**Table 6.** Simulation results for **Scenario 4**.

|  |  | GMAR | TMAR | LMAR | AEP-MAR |
|---|---|---|---|---|---|
| | $\beta_{11}$ | 0.0053 (0.0016) | 0.0063 (0.0019) | 0.0057 (0.0023) | **0.0031 (0.0016)** |
| | $\beta_{12}$ | 0.0050 (0.0015) | 0.0070 (0.0016) | 0.0112 (0.0022) | **−0.0027 (0.0014)** |
| $n = 250$ | $\beta_{21}$ | 0.0116 (0.0023) | 0.0084 (0.0018) | 0.0075 (0.0020) | **0.0055 (0.0015)** |
| | $\beta_{22}$ | 0.0094 (0.0021) | 0.0050 (0.0018) | 0.0105 (0.0016) | **0.0036 (0.0015)** |
| | $\pi_1$ | 0.0067 (0.0023) | −0.0106 (0.0021) | −0.0031 (0.0021) | **−0.0004 (0.0021)** |
| | $\pi_2$ | −0.0067 (0.0023) | 0.0106 (0.0021) | 0.0031 (0.0021) | **0.0004 (0.0021)** |
| | $\beta_{11}$ | 0.0026 (0.0012) | 0.0022 (0.0013) | 0.0010 (0.0019) | **0.0009 (0.0006)** |
| | $\beta_{12}$ | −0.0016 (0.0013) | 0.0046 (0.0014) | 0.0065 (0.0017) | **−0.0015 (0.0010)** |
| $n = 500$ | $\beta_{21}$ | 0.0100 (0.0022) | 0.0030 (0.0013) | 0.0065 (0.0012) | **0.0026 (0.0011)** |
| | $\beta_{22}$ | 0.0086 (0.0022) | −0.0032 (0.0013) | 0.0050 (0.0013) | **0.0023 (0.0013)** |
| | $\pi_1$ | −0.0012 (0.0022) | −0.0171 (0.0023) | −0.0045 (0.0021) | **−0.0012 (0.0020)** |
| | $\pi_2$ | 0.0012 (0.0022) | 0.0171 (0.0023) | 0.0045 (0.0021) | **0.0012 (0.0020)** |

**Table 7.** Simulation results for **Scenario 5**.

|  |  | GMAR | TMAR | LMAR | AEP-MAR |
|---|---|---|---|---|---|
| | $\beta_{11}$ | −0.0149 (0.0079) | −0.1418 (0.0216) | −0.0337 (0.0021) | **−0.0033 (0.0018)** |
| | $\beta_{12}$ | 0.0965 (0.0268) | 0.1251 (0.0170) | 0.0659 (0.0053) | **0.0155 (0.0038)** |
| $n = 250$ | $\beta_{21}$ | −0.0672 (0.0097) | −0.1476 (0.0237) | −0.0761 (0.0068) | **−0.0108 (0.0035)** |
| | $\beta_{22}$ | 0.0173 (0.0264) | −0.1247 (0.0166) | 0.0504 (0.0032) | **0.0075 (0.0029)** |
| | $\pi_1$ | 0.0875 (0.0145) | 0.0577 (0.0054) | 0.0805 (0.0090) | **0.0041 (0.0058)** |
| | $\pi_2$ | −0.0875 (0.0145) | −0.0577 (0.0054) | −0.0805 (0.0090) | **−0.0041 (0.0058)** |
| | $\beta_{11}$ | −0.0218 (0.0030) | −0.1329 (0.0195) | −0.0240 (0.0016) | **−0.0019 (0.0002)** |
| | $\beta_{12}$ | 0.0835 (0.0103) | 0.1183 (0.0151) | 0.0535 (0.0038) | **0.0030 (0.0004)** |
| $n = 500$ | $\beta_{21}$ | −0.0680 (0.0060) | −0.1401 (0.0214) | −0.0704 (0.0056) | **−0.0049 (0.0004)** |
| | $\beta_{22}$ | −0.006 2 (0.0085) | −0.0843 (0.0145) | −0.0466 (0.0025) | **−0.0057 (0.0003)** |
| | $\pi_1$ | 0.0951 (0.0107) | 0.0647 (0.0044) | 0.0800 (0.0076) | **0.0034 (0.0019)** |
| | $\pi_2$ | −0.0951 (0.0107) | −0.0647 (0.0044) | −0.0800 (0.0076) | **−0.0034 (0.0019)** |

**Example 2.** *In this example, we apply numerical simulation to illustrate the finite-sample performance of the model selections for the proposed AEP-MAR model via the Akaike information criterion (AIC) and Bayesian information criterion (BIC). The dataset is generated according to **Scenario 4** in Example 1. We consider the k-component mixture AR model, where $k = 1, 2, 3, 4, 5$. We calculate the AIC and BIC value of the GMAR model, the TMAR model, the LMAR model, and the proposed AEP-MAR model for each k. The corresponding results are shown in Table 8. From Table 8, we find that the the two-component AEP-MAR model is selected by minimizing AIC and BIC.*

**Table 8.** AIC and BIC for **Example 2**.

| | Components | GMAR | | TMAR | | LMAR | | AEP-MAR | |
|---|---|---|---|---|---|---|---|---|---|
| | | **AIC** | **BIC** | **AIC** | **BIC** | **AIC** | **BIC** | **AIC** | **BIC** |
| | 1 | 1498.48 | 1512.54 | 1093.55 | 1195.44 | 1337.81 | 1351.87 | 1024.01 | 1035.10 |
| | 2 | 1420.37 | 1448.48 | 1115.97 | 1154.62 | 1274.51 | 1302.62 | **990.35** | **1032.54** |
| $n = 250$ | 3 | 1109.58 | 1151.74 | 1095.32 | 1155.05 | 1060.34 | 1102.50 | 1004.53 | 1067.77 |
| | 4 | 1257.33 | 1313.54 | 1079.92 | 1160.73 | 1072.44 | 1128.66 | 1014.00 | 1098.32 |
| | 5 | 1003.50 | 1073.77 | 1094.09 | 1195.98 | 1189.40 | 1259.67 | 1000.94 | 1106.34 |
| | 1 | 2028.47 | 2045.31 | 1992.91 | 2013.96 | 1992.99 | 2019.83 | 1981.64 | 2006.90 |
| | 2 | 2172.48 | 2206.16 | 2064.95 | 2111.27 | 2037.35 | 2071.04 | **1944.68** | **1995.21** |
| $n = 500$ | 3 | 2135.74 | 2186.27 | 2132.66 | 2204.24 | 2055.36 | 2105.88 | 2139.67 | 2215.46 |
| | 4 | 2095.95 | 2163.32 | 2120.83 | 2217.68 | 2147.28 | 2214.65 | 2116.72 | 2227.78 |
| | 5 | 2243.17 | 2327.38 | 2112.55 | 2234.66 | 2157.19 | 2241.40 | 2109.66 | 2235.98 |

## 4. A Real Data Analysis

In this section, we will apply the proposed methodology to analyze the daily return series of Hong Kong Hang Seng Index (HSI). The data covers the periods from 2 January 2002 to 31 December 2020, which includes 4689 observations. The original series is shown in Figure 1. From Figure 1, we can clearly see that the daily price series is non-stationary. Similar to [13], we let $x_t = 100 * (\log(P_t) - \log(P_{t-1}))$, where $P_t$ is the daily price in $t$-th day. The corresponding $x_t$ series are shown in Figure 2. We can observe from Figure 2 that $x_t$ is stationary. The skewness and excess kurtosis of $x_t$ are $-0.3$ and $9.01$, respectively, which also means that the $x_t$ series does not satisfy the normality assumption. Meanwhile, the density of log of the daily closing price of the Hong Kong Hang Seng Index is drawn in Figure 3. From Figure 3, we find that the marginal distribution of the series is clearly not symmetric and exhibits multimodality. This indicates that we need to use a mixture AR model rather than an AR model to describe the daily trend of HSI.

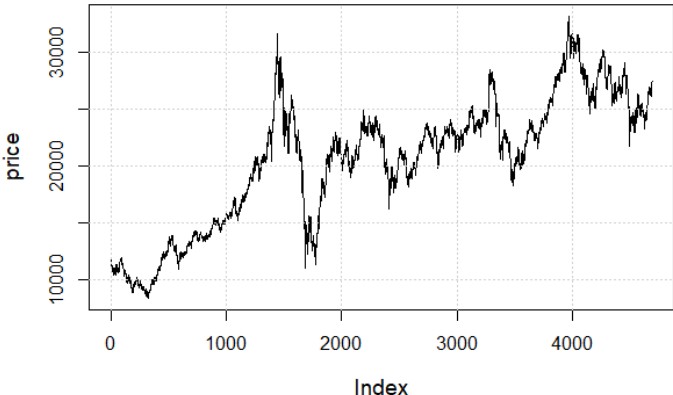

**Figure 1.** The price of Hong Kong Hang Seng Index (HSI), January 2002–December 2020.

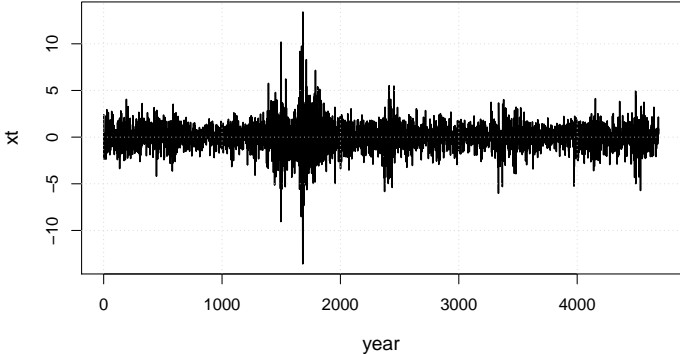

**Figure 2.** The log-return of Hong Kong Hang Seng Index (HSI), January 2002–Dececember 2020.

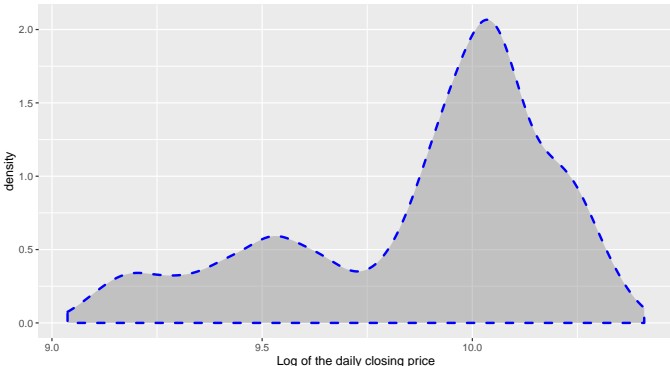

**Figure 3.** Density of log of the daily closing price of Hong Kong Hang Seng Index.

In real data analysis, it is important to choose the number of components *k* and the order of AR components for the MAR model. According to the distribution characteristics described above, we first consider a two-component mixture AR model and a three-component mixture AR model for this dataset. According to Wong & Li (2000) [1], we used AIC and BIC as the model-selection criteria. An example of the performances of the model-selection criteria can be seen in the **Example 2** in Section 3.

The corresponding results are reported in Table 9. From Table 9, we can see that all of the methods rank the two-component as the best. Additionally, the best model selected by minimizing AIC and BIC is the two-component and second-order AEP-MAR model; the value of AIC is 5.61, and the value of BIC is 70.14. This shows that the AEP-MAR model can fit the characteristics of high kurtosis and multimodality in the the log-return series of HSI better. The estimation results for the selected model are given in Table 10. According to Table 10, we obtain the following the two-component and second-order AEP-MAR model for the daily return series of Hong Kong Hang Seng Index.

$$
\begin{aligned}
h(x_t|\mathcal{F}_{t-1},\boldsymbol{\theta}) = \quad & 0.7564 f_1(-0.0447 x_{t-1} + 0.0610 x_{t-2}, 0.5240, 1.2187, 0.4808) \\
& + 0.2436 f_2(-0.6032 x_{t-1} - 0.1394 x_{t-2}, 0.5473, 0.9012, 0.5333),
\end{aligned}
\tag{8}
$$

We obtain from the model (8) that the first component can be interpreted as the overall trend of the log-returns with relatively small fluctuations, and the second component can be interpreted as the irrational "Unilateral Overshooting Phenomenon" in financial markets.

**Table 9.** AIC and BIC of a real dataset.

| Components | Method | AR(2) | | AR(3) | |
|---|---|---|---|---|---|
| | | AIC | BIC | AIC | BIC |
| | GMAR | 5.86 | 70.38 | 9.86 | 87.29 |
| k = 2 | TMAR | 5.79 | 70.32 | 9.80 | 87.23 |
| | LMAR | 5.80 | 70.33 | 9.79 | 87.22 |
| | AEPD-MAR | **5.61** | **70.14** | 9.62 | 87.06 |
| | GMAR | 15.82 | 112.61 | 21.82 | 137.97 |
| k = 3 | TMAR | 15.76 | 112.56 | 21.74 | 137.90 |
| | LMAR | 15.76 | 112.55 | 21.75 | 137.90 |
| | AEPD-MAR | 15.68 | 112.48 | 21.68 | 137.83 |

**Table 10.** The estimation results of two-component and second-order AEP-MAR model.

| Component | $\beta_1$ | $\beta_2$ | $\sigma$ | $\tau$ | $\alpha$ | $\pi$ |
|---|---|---|---|---|---|---|
| component1 | −0.0447 | 0.0610 | 0.5240 | 0.4808 | 1.2187 | 0.7564 |
| component2 | −0.6032 | −0.1394 | 0.5473 | 0.5333 | 0.9012 | 0.2436 |

## 5. Discussion

In this paper, we introduced a robust mixture autoregressive procedure via an asymmetric exponential power distribution. The proposed method has greater flexibility and can adapt to unknown error structures. Under some specific parameters, our method can also be equivalent to the existing method, e.g., GMAR and LMAR as its special cases. In addition, an EM algorithm was introduced to solve the proposed optimization problem. The merits of the proposed method are illustrated by some numerical simulations and a real data analysis. The results indicated that the proposed method was robust and was adaptive to the error distribution. Finally, we will study the large sample properties of the proposed method as future work.

**Author Contributions:** Methodology, Y.J.; Formal analysis, Z.Z. All authors have read and agreed to the published version of the manuscript.

**Funding:** Jiang's research is partially supported by NSFC (12171203) and the Natural Science Foundation of Guangdong (No. 2022A1515010045).

**Institutional Review Board Statement:** Not applicable.

**Informed Consent Statement:** Not applicable.

**Data Availability Statement:** Not applicable.

**Conflicts of Interest:** The authors declare no conflict of interest.

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
