# Peer review of "A Mixture Autoregressive Model Based on an Asymmetric Exponential Power Distribution"

_axioms, doi:10.3390/axioms12020196_

Round 1
Reviewer 1 Report
The manuscript discusses a mixed autoregressive (MAR) model with an asymmetric exponential power (AEP) distribution, which generalizes some of the well-known stochastic distributions. The authors then propose an expectation-maximization (EM) algorithm to estimate the model's parameters, and also present a Monte Carlo simulations for this estimation procedure. Finally, application of AEP-MAR model in describing and analyzing the dynamics of the (multi-modal) distribution of the dynamics of the log-return series of the Hong Kong Hang Seng Index was presented.
The topic and idea of the manuscript are interesting and (relatively) new, and it is written in a (relatively) good and precise way. Therefore, I think it should be taken into consideration. However, my main objections refer, first of all, to the scope and content of the manuscript, which seems very incomplete. Here are some suggestions:
1. The authors, as I have already pointed out, in a quite satisfactory way describe the new, so-called AEP-MAR model. However, in the "Methodology" section, more words should be devoted to the stochastic properties of the model itself, i.e. one should, first of all, describe its moments, autocorrelation, etc.
2. The EM-algorithm is solidly explained, but a few words should be added here, first of all, regarding its convergence and asymptotic properties.
3. The most interesting (but also the most problematic) part of the manuscript refers to Sect. 3, where the simulation of the AEP-MAR model is discussed and compared with similar models of this type. Here, first of all, I am not sure if the AEP-MAR model is used as a generalization of other models, i.e. whether the noise-distributions used here are special cases of the AEP-distribution or are they taken arbitrarily. In the second case, I don't see why the AEP-MAR model, say, would be used to evaluate MAR-models with Student's or chi-square innovations. In any case, the authors should explain to the reader in more detail what their idea was with the choice of distributions and models shown. On the other hand, the comparison and verification of efficiency should be possible with another indicators, such as bias, relative error, etc., and not exclusively by applying the mean squared error (MSE). Finally, it can be seen from the results themselves that the estimators, especially on a small sample, are very imprecise, and the reason for this may be the small number of (only) 100 replications. Perhaps the authors took such a small number of replications due to the large computational time and relative complexity of the models themselves and the EM-procedure of parameter estimation? However, if it is feasible, I think that they should be taken in larger numbers of replications.
4. Finally, part of the manuscript in Sect. 4, which refers to the application of the AEP-MAR model, is surprisingly short and very "sketchy". Here, the authors should first describe in detail, statistically, the observed set of empirical data, especially its multimodal structure. (First of all, as the authors themselves say: "the number of components k and the order of AR components according to AIC and BIC..") Also, the mathematical appearance of the obtained model should be shown, as well as the quality of its fitting in the way it is done in the simulation part. To this end, a comparison with previously known MAR-models would also be quite appropriate here.
Minor comments:
Line 44: Please change the beginning of the sentence "The rest of the paper..." to a more appropriate one, and also "a"-> "an".
Line 71: "We"-> "we".
Author Response
The response to Reviewer 1 is given in the attachment.

Reviewer 2 Report
I have read the paper, and i have some comments
1- Why did the authors use these distributions in the mixture?
2- The abstract is too small and does not reflect all the work; it should be extended
3-The first paragraph of the introduction should be rewritten
4-The first paragraph of the Methodology should be rewritten
5-The conclusion and the discussion are not well written
I suggest rewriting them.
After all, the paper needs some modifications, suggest a major revision
Author Response
The response to Reviewer 2 is given in the attachment.

Reviewer 3 Report
I think the paper is interesting and, in general, worth publishing. I have, however, some comments that can be useful in preparing the final version of the manuscript. It would be expected the Authors could address them
1. Please, clarify in the abstract and introduction what is the main contribution of the paper and its main novelty in comparing to the existing literature.
2. Please, justify and explain that the useness of assymetric exponential power distribution in autoregressive model is indeed motivated by the practice or time series theory.
3. I recommend a detailed proofreading of the whole manuscript to avoid typos and linguistic errors. Below I give some examples which should be corrected.
Examples of typos and liguistic errors:
(1) Abstract, line 2: change "which including" by "includes" or "including";
(2) Abstract, line 7: replace "our proposed method" by "our method" or "the proposed method";
(3) page 9, line 9: change "student t-mixture" into "Student t-mixture".
Author Response
The response to Reviewer 3 is given in the attachment.

Round 2
Reviewer 1 Report
In this version, the authors have (slightly) modified the manuscript based on my suggestions. However, the corrections are minor, while, on the other hand, their answers to my remarks are very detailed and, above all, correctly worded. Therefore, I suggest the authors make one more effort regarding the following:
1. In Sec. 3 where the simulations of their model are described, it would be good if the detailed response addressed to me was, to some extent, included in the manuscript itself. Thus, as I pointed out earlier, the reader would be completely familiar with the idea of this way of simulating a new model. In addition, although the authors claim this in the report notes, I did not notice that any additional checks on the effectiveness of the obtained ratings were used here.
2. Unfortunately, in Sec. 4 additions are only insignificant and it is (still) quite short and incomplete. I will ask the authors to once again carefully study my suggestions from the previous review and expand this part of the manuscript in adequate ways.
Author Response
The reponse is given in the attachment.

Round 3
Reviewer 1 Report
In this version of the manuscript, the authors have (finally) fulfilled all my remarks and suggestions. Now the manuscript really seems complete as a whole, and I believe that it should be accepted.